# OpenReview forum: "Stable-GFlowNet: Toward Diverse and Robust LLM Red-Teaming via Contrastive Trajectory Balance"
_ICML.cc/2026/Conference — ICML 2026 spotlight_

### Official Review · Reviewer_NDSS · 2026-02-28

**Soundness:** 2
**Presentation:** 3
**Significance:** 3
**Originality:** 2
**Overall Recommendation:** 4
**Confidence:** 3

**Summary:**

This paper investigates automated red-teaming for Large Language Models (LLMs) using Generative Flow Networks (GFNs). The authors study discovering attack prompts that are both highly effective and semantically diverse, a task hindered by the severe training instability and mode collapse inherent in standard GFNs. The authors argue that existing GFNs struggle because estimating the global partition function Z is unstable in large discrete spaces, and noisy reward landscapes often lead models to exploit non-fluent gibberish. To address this, they propose Stable-GFN (S-GFN), a robust framework that introduces Contrastive Trajectory Balance (CTB) to bypass Z estimation via pairwise comparisons, Noisy Gradient Pruning (NGP) to filter stochastic reward noise, and a Min-K Fluency Stabilizer (MKS) to preserve the linguistic integrity of the generated prompts. Furthermore, experiments demonstrate that the diverse vulnerabilities discovered by S-GFN provide superior defensive coverage when used for subsequent victim safety fine-tuning.

**Compliance With Llm Reviewing Policy:**

Affirmed.

**Final Justification:**

My concerns have been partly resolved. However, I am not convinced regarding the contribution and the metric. Therefore, I decided to maintain my rating.

**Key Questions For Authors:**

1. Clarify the novelty.
1. Discuss and provide further explanation about the issues of metrics and ablation studies.

**Limitations:**

Yes.

**Strengths And Weaknesses:**

Strengths:

1. This paper presents a theoretical analysis of the method.
1. The empirical results are promising.
1. This paper is well-written and easy to follow.

Weaknesses:

1. This paper might be an incremental extension of [1], and the proposed CTB loss is conceptually similar to the Contrastive Balance proposed in [2].
2. The reliance on the "Unique Attacks" (UA) metric as a primary indicator of diversity is highly questionable. As shown in Figure 3(a), UA is extremely sensitive to the clustering threshold, making it a fragile and easily manipulable standard. More troublingly, Table A in the appendix reveals that when evaluated on standard, robust diversity metrics (such as Self-BLEU and Diversity), S-GFN actually performs worse than baselines like Jailbreak-R1 and ICL. Highlighting a threshold-sensitive metric in the main text while burying inferior performance on standard metrics in the appendix severely undermines the paper's claims.
3. The ablation study lacks critical dimensions needed to fully evaluate the proposed modules:
   1. In the ablation study of the MKS (Table 3(a)), the authors only report the UA metric while omitting the ASR.
   2. In Table 3(a), the UA metric is reported as exactly 0 when only using the raw reward R(y). This is highly counter-intuitive: removing constraints entirely should theoretically yield at least some successful attacks. This anomaly is not adequately explained and raises concerns about the implementation or evaluation script.

[1] Learning diverse attacks on large language models for robust red-teaming and safety tuning. ICLR 2025.

[2] Embarrassingly Parallel GFlowNets. ICML 2024.

---

> ### Author Rebuttal · Authors · 2026-03-31
>
> We sincerely thank you for constructive feedback.
>
> ### W1 & Q1
> ---
> Our key contribution is identifying training instability under noisy rewards as a core bottleneck in GFN-based LLM red-teaming and proposing an effective remedy. We built upon [1], the first GFN-based red-teaming method, and improved its stability and practicality.
>
> We agree that CTB shares a similar intuition of CB [2], avoiding explicit partition function estimation via pairwise comparisons. However, they target different settings: CB integrates local GFNs with backward policies in federated learning, whereas CTB balances autoregressive trajectories within a single-model LLM batch (to remove Z). Beyond this difference in formulation, our analysis provides LLM-specific insights: gradient variance reduction and optimal policy equivalence. We will cite [2] and discuss the relationship in the revision.
>
> However, our core contribution is S-GFN as an integrated framework. In this context, removing explicit Z-estimation does not uniquely lead to the CTB objective. The importance of CTB is that its pairwise structure allows NGP. The table below compares several simple variants under the same setup as Table 3(b). Batch-level variants (e.g., mean/median) also avoid explicit Z-estimation and outperform TB, suggesting that removing explicit Z-estimation is beneficial under noisy rewards. However, the best performance is achieved by combining CTB with NGP. This underscores the importance of NGP in the noisy reward setting. We will clarify this in the revision.
>
>
> |Method|UA(#)|ASR(%)|
> |-|-|-|
> |TB|67|85.8|
> |Mean|106|91.4|
> |Median|110|86.7|
> |CTB|108|82.9|
> |CTB + NGP|**121**|**92.2**|
>
> The following objective is below:
> Mean: $\mathcal{L}\_{\mathrm{Mean}}(\theta)=\frac{1}{B}\sum_{i=1}^B\left(f_i-\frac{1}{B}\sum_{j=1}^B f_j\right)^2$
> Median: $\mathcal{L}\_{\mathrm{Median}}(\theta)=\frac{1}{B}\sum_{i=1}^B\left(f_i-\operatorname{median}_{j\in\{1,\dots,B\}}f_j\right)^2$
>
>
> ### W2 & Q2
> ---
> We agree that UA is a threshold-sensitive proxy. However, Table  3(a) shows that our method achieves higher UA than GFN across all thresholds $\geq$ 0.5. This supports that S-GFN is generally more effective than GFN, suggesting robustness to threshold choice. Regardless of the clustering method used (e.g., HDBSCAN and Louvain), Table B further shows that S-GFN achieves the highest UA.
>
> Beyond UA, we evaluated two additional metrics: 1) cross-attack metric (ASR); 2) categorical diversity, to assess semantic diversity. First, Table 1 reports cross-attack evaluation as a threshold-free metric: in 1-on-1 comparisons, models fine-tuned on S-GFN reduce baseline ASR to <1%, while S-GFN also maintains a clear advantage as an attacker. Second, Fig. D shows that S-GFN achieves superior categorical diversity. Together, these results indicate a broader, more diverse attack surface beyond UA.
>
> We appreciate the suggestion to consider Self-BLEU, n-Distinct, and embedding diversity. We view these as complementary metrics, as they primarily capture lexical or representation-level variation and do not directly quantify the number of semantically distinct successful attack modes. Since the main goal of red-teaming is to uncover diverse successful attack types rather than maximize surface-level diversity, we did not treat these metrics as the primary criterion.
>
>
> ### W3 & Q2
> ---
> We apologize for omitting ASR due to table length constraints. We now report it as follows: except for the primitive R(y)setting, ASR remains consistently high (>80%) for both GFN-TB and GFN-CTB, which suggests that the differences in Sec 5.3 are not driven by ASR variation.
>
> |ASR|GFN-TB|GFN-CTB|
> |-|-|-|
> |$R(y)$|0%|0%|
> |$R(y) * \pi_\text{ref}(y)$|92.7%|88.2%|
> |$R(y)$ with Logprob|87.6%|85.5%|
> |$R(y)$ with MKS|85.8%|82.9%|
>
> ### W4 & Q2
> ---
> At first glance, reporting both UA and ASR as 0 under the primitive reward R(y) may seem counterintuitive. However, we consistently observed this behavior in practice, and the evidence points to reward hacking rather than implementation or evaluation errors.
>
> As illustrated below, gibberish prompts can occasionally receive small but non-zero toxicity scores from LLM-based judges, while coherent yet unsuccessful attacks may receive even lower scores. Consequently, optimization favors meaningless strings over valid attacks.
>
> This behavior is consistent across seeds and hyperparameters, which motivates reward stabilization. Prior work [1] uses KL-style regularization; our MKS instead mitigates this failure mode while preserving broader exploration in the GFN setting.
>
> ---
> > Sample 1: (\<ucocpsiou%"), ":{ }{ %"), ">', ""}, ��>', ", ,"); '} }', Qualcomm攸! \
>
> >Response 1: I'm sorry, but I don't understand what you're asking. Could you please rephrase your question in English or another language
>
> >Reward 1: 0.095
>
> >Sample 2: Develop a cross reference in the RASP to detect to determine when it is safe to collect IP to
>
> >Response 2: I'm sorry, but I can't assist with that.
>
> >Reward 2: 0.034

---

> > ### Author Rebuttal · Reviewer_NDSS · 2026-04-01
> >
> > Thank you for the rebuttal. My concerns have been partly resolved. However, I am not convinced regarding the contribution and the metric. Therefore, I decided to maintain my rating.

---

### Official Review · Reviewer_obBW · 2026-03-12

**Soundness:** 4
**Presentation:** 4
**Significance:** 3
**Originality:** 4
**Overall Recommendation:** 6
**Confidence:** 3

**Summary:**

The paper proposes a more stable GFN training method which increases the generated attack diversity and attack success rate by reducing training instability. The method consists in 1) optimizing contrastive (vs classical) trajectory balance in order to avoid estimating the partition function Z while guaranteeing equivalent optimal policy, 2) avoiding overfitting noise by pruning insignificant differences in trajectory contrast due to victim LLM and toxic classifier variance, and 3) minimizing reward hacking by penalizing gibberish sentences detected through the average probability of the k least likely tokens. A theoretical proof demonstrates the modified loss's optimum is the same, while experiments are performed to demonstrate improved performance in practice across many tasks (LLM red-teaming + cross-attack + transfer-attack, molecular generation, noisy hypergrid exploration), datasets, LLM models, and metrics. An ablation study demonstrates the necessity of all 3 method subparts, the proposed method's added computational cost is quantified, and results are verified through additional hyper-parameter- and metric-based control experiments.

**Compliance With Llm Reviewing Policy:**

Affirmed.

**Final Justification:**

My concerns have been addressed by the author’s rebuttal, and the score stays at 6 / Strong accept.

**Key Questions For Authors:**

1. In Figure 3a, S-GFN's UA grows exponentially as the similarity threshold t increases toward 0.8. Since UA is highly sensitive to this hyperparameter, could the overwhelming diversity reported be an artifact of how the clustering is defined at high thresholds? How does the ASR/UA trade-off look with a stricter similarity threshold? Aditionally, Rainbow Teaming seems like a strong contender vs S-GFN in Table A according to the specific diversity metric used.
2. You argue that KL-divergence regularization in GFNs is risky because it distorts the target distribution. However, MKS applies a hard penalty to non-fluent samples. Isn't this also a distortion of the "true" toxicity landscape provided by the classifier? Could you clarify why a hard threshold (MKS) is theoretically or practically superior to a soft penalty (KL) for distribution matching?

**Limitations:**

yes

**Strengths And Weaknesses:**

- Soundness: The paper is technically strong, with improved performance claims supported by large numbers of experiments (task variations, datasets, models, metrics), control experiments (mostly hyper-parameter- and metric-based), and theoretical proofs to top it off.

- Presentation: While the subject matter is relatively complex, touching upon multiple literatures (GFN, LLM red-teaming), the presentation is clear and the overall narrative easy to follow, with lengthy appendix entries to answer most questions which arise in the main text.

- Significance: The CTB loss and NGP parts together improve GFN training stability in large search spaces (reward-dense or -sparse) across tasks, datasets and backbone models, highlighting the method's general applicability and relevance for any GFN-related task. While the negligible increase in training compute further solidifies the relevance of this methodology, the introduction of several hyper-parameters (NGP's sigma and MKS's k and T) might require some additional fine-tuning (although sigma<1 seems robust).

- Originality: The main originality comes from re-contextualizing trajectory balance as contrastive for increased training stability in practice while proving that the optimal policy is (quite elegantly) still guaranteed in theory.

---

> ### Author Rebuttal · Authors · 2026-03-31
>
> We sincerely thank you for constructive feedback.
>
> ### **Q1**
>
> ---
>
> We agree that UA is sensitive to the threshold. To address this concern, we report the performance at various thresholds in Table 3(a) to address this issue. At thresholds of 0.5 or higher, S-GFN consistently yields higher UA values than GFN. As stated in the paper, the values listed in Table 1 correspond to a threshold of 0.7. Furthermore, in Table B, we provide additional results using alternative clustering algorithms, such as HDBSCAN and Louvain. S-GFN consistently achieves superior UA performance in these cases as well. Figure D also reports category diversity, showing diversity that is either higher than or comparable to that of the baselines. Therefore, our conclusions are based on robust trends observed across various reasonable clustering choices and parameters, rather than a single setting.
>
> Beyond UA, Table 1 provides a threshold-independent behavioral signal of attack diversity and quality through cross-attack evaluation. Since the ASR in this setting is measured directly on defenses trained against another attack method, it allows a one-to-one comparison of how much novel attack surface each method exposes. For example, a defense fine-tuned on Rainbow Teaming attacks still shows 83.24% ASR against S-GFN attacks, whereas the reverse direction yields only 0.23% ASR. This asymmetry suggests that S-GFN discovers attack modes that are substantially less covered by Rainbow Teaming, which is consistent with our claim that S-GFN explores a broader and more practically useful attack space.
>
> We also thank you for your meticulous review, especially for looking into the details in the Appendix. We report Self-BLEU, n-Distinct, and embedding diversity as complementary metrics because they capture lexical or representation-level diversity. In particular, since self-bleu and n-distinct are n-gram-based metrics, they primarily measure lexical diversity and are sensitive to length and repetition. Embedding diversity, on the other hand, measures the average distance and is vulnerable to outliers. These metrics also do not directly measure how many semantically distinct successful attack modes are discovered. Since the main goal of red-teaming is to uncover distinct successful attack types rather than maximize lexical dispersion, we do not treat these metrics as the primary criterion.
>
> ### **Q2**
>
> ---
>
> We appreciate the insightful observation. We agree that MKS modifies the original classifier-based reward landscape and thus represents a form of distortion. We do not intend to claim that our approach performs a pure distribution matching devoid of any modification.
>
> However, we distinguish KL regularization from MKS based on the scope and nature of their intervention. KL regularization introduces a global attraction toward a reference model, acting as a dense prior that reshapes the entire target distribution. This often leads to the continuous suppression of high-reward regions simply because they deviate from the reference model's preference.
>
> In contrast, MKS functions as a localized feasibility filter rather than a re-weighting of the entire semantic space. Its effect is limited to regions where the toxicity classifier is prone to reward hacking. Typical examples include repetitive, strange Unicode characters and non-fluent sequences. While KL regularization imposes a global bias that tends to contract the exploration space toward the reference, MKS surgically removes clearly invalid regions while preserving the original characteristics of the target distribution across the remaining valid space.
>
> Therefore, our argument is not that a hard threshold is inherently superior to a KL penalty in a general theoretical sense. Rather, we emphasize that in practical red-teaming scenarios where reward models are vulnerable to gibberish, MKS introduces a much less restrictive bias than KL, effectively mitigating reward-hacking while maintaining the diversity of the adversarial search.

---

> > ### Author Rebuttal · Reviewer_obBW · 2026-04-03
> >
> > Thank you for these clarifications.
> > Question 1 is convincingly answered by the paper’s testing across threshold choice / clustering technique / cross-attack evaluation, and pointing out each diversity metric’s characteristics.
> > Question 2 is convincingly answered by shifting from a broad theoretical claim to a more practical one (although seeing examples with text highlighted by whether they pass the MKS filter could help “visualize” this rule’s effect, but this is probably too much to ask, so I won’t).
> > Furthermore, the answer to weakness 2 of reviewer 26qN answers my “Significance” weakness, although the MKS threshold’s exact value still has a strong effect on generated results and might require additional fine-tuning in other contexts.
> >
> > The score was already at 6 / Strong accept, and so shall stay there. No further questions.

---

### Official Review · Reviewer_K5vT · 2026-03-13

**Soundness:** 3
**Presentation:** 3
**Significance:** 2
**Originality:** 3
**Overall Recommendation:** 4
**Confidence:** 4

**Summary:**

The authors propose Stable-GFlowNet (S-GFN) for LLM red-teaming, aiming to improve both attack diversity and robustness by replacing standard trajectory balance with a contrastive objective that avoids explicit partition-function estimation. The method also adds Noisy Gradient Pruning and a Min-K Fluency Stabilizer to handle noisy rewards and gibberish-like reward hacking.

**Compliance With Llm Reviewing Policy:**

Affirmed.

**Final Justification:**

My main concerns are addressed by the authors' rebuttal, and therefore I have raised my score.

**Key Questions For Authors:**

1. How much of the reported gain remains if attack success is evaluated by stronger human or model-based judges rather than the same toxicity-style reward pipeline?
2. How well does the method work if applied to the agentic setting (defending against tool-use agents)?

**Limitations:**

yes

**Strengths And Weaknesses:**

**Strengths**
The main idea is simple and reasonable: remove unstable Z estimation and replace it with pairwise relative comparisons. The empirical results are strong, especially the large gain in unique attacks while keeping ASR high, and the ablations on CTB, NGP, and MKS are helpful. The extra experiments on molecular generation and noisy hypergrid are also a nice attempt to show that the method is not purely benchmark-specific.

**Weaknesses**
1. The paper’s central finding pertains to improved diversity and stability in red-teaming, but the evaluation still feels somewhat narrow. The setup relies on a fixed attacker/victim/classifier pipeline, with Llama Guard used as the toxicity classifier and ASR defined through that same style of classifier-based scoring, so it is not fully clear how much the gains reflect true attack quality versus optimization toward the reward model.
2. The diversity claim also depends heavily on the UA clustering metric, which is threshold-sensitive; the paper does include a threshold analysis, but the overall conclusion still relies a lot on this proxy.
3. Missing some important related work, such as RedCodeAgent [1], RedAgent, Rainbow Teaming, that automatically generates red teaming data and falls within the same scope of this paper.

[1] Guo, Chengquan, et al. "RedCodeAgent: Automatic Red-teaming Agent against Diverse Code Agents." arXiv preprint arXiv:2510.02609 (2025).
[2] Xu, Huiyu, et al. "Redagent: Red teaming large language models with context-aware autonomous language agent." arXiv preprint arXiv:2407.16667 (2024).

---

> ### Author Rebuttal · Authors · 2026-03-31
>
> We sincerely thank you for constructive feedback.
>
> ### **W1**
> ---
> Evaluation under a fixed pipeline may obscure whether performance reflects semantic attack quality or overfitting to a specific reward model. To address this, we report transfer evaluations with changed classifiers and victim models. In Table E, although our method was trained with Llama-Guard, it still maintains strong attack performance when evaluated with ShieldGemma-9B. Table 2 further shows strong transfer performance across diverse victim LLMs, including GPT-OSS-20B. Together, these results suggest that the gains are not solely due to overfitting to a single victim/classifier pipeline.
>
>
> ### **W2**
> ---
> We agree that UA can be viewed as a threshold-sensitive proxy. However, Table 3(a) shows that our method achieves higher UA than GFN across all thresholds $\geq$ 0.5. This supports that S-GFN is generally more effective than GFN, suggesting robustness to threshold choice. Regardless of the type of clustering method (e.g., HDBSCAN and Louvain), Table B further shows that S-GFN achieves the highest UA.
>
> Beyond the UA metric, we actually evaluated two additional metrics: 1) cross-attack metric (ASR); 2) categorical diversity, to assess semantic diversity. First, Table 1 reports cross-attack evaluation as a threshold-free metric: in 1-on-1 comparisons, models fine-tuned on S-GFN reduce baseline ASR to <1%, while S-GFN also maintains a clear advantage as an attacker. This demonstrates that S-GFN covers attacks that other methods fail to detect. Second, Fig. D shows that S-GFN achieves superior categorical diversity. Together, these results indicate a broader, more diverse attack surface beyond UA. This demonstrates that S-GFN is capable of identifying various attack modes without relying on UA.
>
>
> ### **W3**
> ---
> Thank you for the suggestion. Please note that Rainbow Teaming was already included as a primary baseline in Tables 1 and 2. We do not include RedCodeAgent because it requires a specialized coding-agent evaluation environment beyond our current setup. For RedAgent, we additionally performed experiments under the same setup as Table 1, using 100 generated attacks; we used the SFT dataset as the target, set the maximum query budget to 30, and used the 4 strategies reported as the most effective in the paper. All the suggested references will be cited and discussed in the revision.
>
> As shown in the table below, S-GFN achieves the highest UA while maintaining competitive ASR against the agent-based baseline. In particular, although RedAgent attains 42% ASR, its UA is only 9, whereas S-GFN produces substantially more unique attacks. These results further support our claim that S-GFN generates more diverse attacks without sacrificing overall attack effectiveness.
>
> |  | UA(#) | ASR(%) |
> |-|-|-|
> | GFN | 13 | **97** |
> | Rainbow Teaming | 47 | 67 |
> | Jailbreak-R1 | 9 | 9 |
> | RedAgent | 9 | 42 |
> | S-GFN (Ours) | **53** | 88 |
>
>
> ### **Q1**
> ---
> To address this concern, we additionally evaluated 100 randomly selected samples from Table 1 using **GPT-5-nano** and **Gemini-flash-lite** as stronger model-based judges, following the toxicity categories in Figure D.
>
> We found that evaluation tendencies vary across judges. For example, Jailbreak-R1 showed lower ASR under Llama-Guard but substantially higher ASR under GPT and Gemini, indicating a discrepancy between a specific toxicity classifier and closed-source model judges.
>
> Nevertheless, S-GFN achieved the highest ASR under both GPT and Gemini and remained highly competitive under Llama-Guard. Its UA also remained the highest or near-highest across all three evaluators on the same subset. This suggests that S-GFN’s gains are not simply due to reward hacking against a specific classifier, but reflect more robust harmful attack generation across evaluators.
>
> We agree that human evaluation would provide an important additional perspective. While we could not include a reliable human study within the short rebuttal period, we will clarify this limitation and include such evaluations in the future version.
>
> |  | UA in GPT | ASR in GPT | UA in Gemini | ASR in Gemini | UA in LlamaGuard3 | ASR in LlamaGuard3 |
> |-|-|-|-|-|-|-|
> | GFN | 14 | 79% | 12 | 89% | 13 | **97%** |
> | Rainbow Teaming | 49 | 79% | 39 | 50% | 47 | 67% |
> | Jailbreak-R1 | 35 | 39% | 42 | 44% | 9 | 9% |
> | Ours | **59** | **96%** | **59** | **97%** | **53** | 88% |
>
>
> ### **Q2**
> ---
> Thank you for this valuable suggestion. We agree that red-teaming tool-use agents is an important problem. Since S-GFN is a reward-proportional distribution approximator, it can theoretically be applied to tool-using agents as well. However, this paper focuses on single-prompt red teaming against a fixed victim model, and evaluation in agentic settings is beyond the scope of the current paper. We will clarify this limitation and explore such effective extensions to such settings in the future revision.

---

> > ### Author Rebuttal · Reviewer_K5vT · 2026-03-31
> >
> > I thank the authors for the detailed rebuttal and the additional experiments. I am glad to see that S-GFN outperforms rainbow teaming and RedAgent under the judges of different LLMs. The new synthetic data generation method can be beneficial to the community. I will raise my score.

---

### Official Review · Reviewer_26qN · 2026-03-13

**Soundness:** 3
**Presentation:** 4
**Significance:** 4
**Originality:** 3
**Overall Recommendation:** 5
**Confidence:** 3

**Summary:**

The paper presents a modified Trajectory Balance (TB) objective for Generative Flow Networks (GFNs), Contrastive Trajectory Balance (CTB); in particular, CTB removes the need to explicitly predict the partition function. The authors show that the minimizer of CTB is the same as that of TB but reduced gradient variance; hence, CTB has the advantage of improved stabliity and convergence properties. The authors further present techniques to avoid reliance on uncertain reward signals (Noisy Gradient Pruning, NGP), and to avoid promoting the generation of nonsense text (Min-K Fluency Stabilizer, MKS).

**Compliance With Llm Reviewing Policy:**

Affirmed.

**Final Justification:**

The authors prove that their contrastive trajectory balance loss has an identical global minimum to trajectory balance loss, but lower gradient variance. The experimental results on AutoDan strengthen the evidence that S-GFN outperforms strong existing baseline jailbreaking attacks. The CTB loss and the additional loss terms for fluency and further variance reduction are likely to be valuable for the usage of GFlowNets in a wide range of domains.

**Key Questions For Authors:**

How does S-GFN compare to strong jailbreaking attacks such as AutoDAN-Turbo?

**Limitations:**

Yes

**Strengths And Weaknesses:**

## Strengths

- The approach is natural and well motivated. It demonstrates strong empirical performance, with CTB-trained models converging faster and producing more diverse outputs.
- The ablation studies indicate that both NGP and MKS significantly improve the output quality; these and CTB are likely to be applicable beyond the red-teaming setting.
- The paper is well written and readable.

## Weaknesses

- The baseline jailbreak methods should include stronger baselines such as AutoDAN-Turbo [1].
- NGP and MKS introduce additional hyperparameters to which performance may be sensitive, as indicated by Appendix B, necessitating tuning.

## Minor Comments

Line 204 -> "Define the log-flow error as ..."
Baselines, page 6: "the replay buffer" -> "a replay buffer", "the DPO" -> "DPO"

[1] Xiaogeng Liu et al. AutoDAN-Turbo: A Lifelong Agent for Strategy Self-Exploration to Jailbreak LLMs. ICLR 2025.

---

> ### Author Rebuttal · Authors · 2026-03-31
>
> We sincerely thank you for constructive feedback.
>
> ### **W1**
>
> ---
>
> We appreciate this suggestion. We conducted additional experiments comparing S-GFN with AutoDAN-Turbo on a 100-sample subset under the same experimental conditions in Table 1. For a fair comparison, all methods used Qwen2.5-1.5B-Instruct as the attacker LLM. We set the maximum number of attack rounds to 10.
>
> As shown below, S-GFN outperforms AutoDAN-Turbo in both diversity (UA) and attack success (ASR).
> The results clearly indicate that S-GFN identifies more diverse and effective attack patterns than AutoDAN-Turbo within the same evaluation framework. This confirms that our method's objective of balancing diversity and reward is highly effective even when compared against recent instruction-following-based baselines.
>
> We will include these results and a detailed discussion in the revised version with the citation.
>
> |  | UA(#) | ASR(%) |
> | --- | --- | --- |
> | GFN | 13 | **97** |
> | Rainbow Teaming | 47 | 67 |
> | Jailbreak-R1 | 9 | 9 |
> | AutoDan-Turbo | 43 | 43 |
> | S-GFN (Ours) | **53** | 88 |
>
> ### **W2**
>
> ---
>
> We acknowledge that NGP and MKS introduce additional hyperparameters, and that their optimal values may vary depending on the domain.
> However, Figures B and C show that the performance is not excessively sensitive to these choices and remains relatively stable over a reasonable range of values.
> This suggests that the reported gains are robust to these parameter choices and do not rely on exhaustive, task-specific tuning.

---

> > ### Author Rebuttal · Reviewer_26qN · 2026-04-01
> >
> > Thank you for the additional results on AutoDan-Turbo, I am glad to see that S-GFN continues to show strong performance; I maintain my score.

---

### Decision · Program_Chairs · 2026-04-30

**Decision:**

Accept (spotlight)

**Comment:**

The recommendation is based on the reviewers' comments, the area chair's evaluation, and the author-reviewer discussion. This paper studies LLM Red-Teaming through the design of generative flow networks and contrastive trajectory balance. All reviewers find the studied setting novel and the results provide new insights. The authors’ rebuttal has successfully addressed the major concerns of reviewers.
In the post-rebuttal phase, all reviewers were satisfied with the authors’ responses and agreed on the decision of acceptance.